# Understanding Stunting: Impact, Causes, and Strategy to Accelerate Stunting Reduction—A Narrative Review

**DOI:** 10.3390/nu17091493

**Published:** 2025-04-29

**Authors:** Aisyah Tri Mulyani, Miski Aghnia Khairinisa, Alfi Khatib, Anis Yohana Chaerunisaa

**Affiliations:** 1Magister Program, Faculty of Pharmacy, University of Padjadjaran, Jalan Raya Bandung Sumedang km 21 Jatinangor, Sumedang 45363, Indonesia; aisyah17015@mail.unpad.ac.id; 2Department of Pharmacology and Clinical Pharmacy, Faculty of Pharmacy, University of Padjadjaran, Jalan Raya Bandung Sumedang km 21 Jatinangor, Sumedang 45363, Indonesia; miski.aghnia@unpad.ac.id; 3Department of Pharmaceutical Chemistry, Faculty of Pharmacy, International Islamic University Malaysia (IIUM), Kuantan 25200, Pahang, Malaysia; alfikhatib@iium.edu.my; 4Department of Pharmaceutics and Pharmaceutical Technology, Faculty of Pharmacy, University of Padjadjaran, Jalan Raya Bandung Sumedang km 21 Jatinangor, Sumedang 45363, Indonesia

**Keywords:** children, dysbiosis, growth failure, stunting, undernutrition

## Abstract

Stunting is a major global health concern, particularly in low- and middle-income countries, due to its persistently high prevalence. It often originates from chronic malnutrition during the critical first 1000 days of life. Maternal and child nutrition are critical determinants of a child’s growth and development. This article aimed to explore the impact, causes, and evidence-based strategies to accelerate the reduction of stunting incidence worldwide. This review was undertaken with sources from PubMed, Scopus, Google Scholar, Science Direct, and MEDLINE from October 2024 to January 2025. This review was undertaken with sources from PubMed, Scopus, Google Scholar, Science Direct, and MEDLINE from October 2024 to January 2025 using the keyword “Stunting”, “Causes of stunting”, “Stunting Impact”, “Stunting Intervention”, and “Stunting Prevention”. The findings highlight the multifactorial causes of stunting, including maternal malnutrition, inadequate breastfeeding and complementary feeding, poor sanitation, and socioeconomic factors. Stunting is associated with impaired linear growth, cognitive deficits, gut dysbiosis, endocrine disruption, anemia, and increased risk of chronic diseases later in life. Addressing stunting demands multisectoral strategies focusing on maternal and child nutrition, infection prevention, improved WASH (Water, Sanitation, and Hygiene) practices, and socioeconomic support. The evidence presented may guide policy development and targeted interventions to prevent stunting and its long-term effects.

## 1. Introduction

Stunting can be described as a condition where a child is significantly shorter than the expected height for their age. The WHO has a standard for measuring stunting in children, which is measured by the “Z score” assessment [1,2]. Stunting is a condition where a child’s height-to-age ratio is less than two standard deviations (SDs) below the median value for their age, indicating potential limitations in the child’s ongoing development. Globally, it is agreed to identify stunting by measuring the body length ratio of −2 SD against the age from the WHO median standard value, which is categorized as mild or moderate stunting. Meanwhile, children with a body length ratio of −3 SD against age from the WHO median standard value are categorized as having severe stunting [2,3]. Stunting in children begins from the early stages of life until the first three or four years [1,4,5]. The first 1000 days of life are a crucial period for determining a child’s future development, both in terms of physical and mental health [6,7,8]. Stunting syndrome can be identified as part of a continuous process cycle, starting from maternal nutrition during pregnancy, which is passed on to the child and continues into the next cycle through several factors, as explained in Figure 1 [9].

The causes of stunting, according to the conceptual framework proposed by the WHO, represent a complex and multifactorial issue. These factors include household conditions, inadequate complementary feeding, non-exclusive breastfeeding, infectious diseases, political economy, education, access to healthcare, social and cultural influences, environmental conditions, sanitation, and water quality [10]. Infant development is influenced by maternal nutrition, starting from pregnancy and continuing throughout breastfeeding. The WHO recommends exclusive breastfeeding until 2 years of age, followed by the gradual introduction of complementary foods starting at 6 months of age [11,12]. After reaching 6 months of age, children’s energy and nutritional needs increase, so in addition to breastfeeding, complementary feeding is necessary to meet these additional energy and nutritional needs [13]. Poor maternal nutrition during pregnancy can limit fetal development and increase the consequences of stunting in children [14]. Inadequate complementary feeding involves insufficient food, excessive reliance on instant foods (often only mixed with water), using contaminated water, and storing food improperly in open spaces or unclean containers, allowing bacteria to grow and potentially leading to infections in children [10]. In addition to inadequate nutrition, poor sanitation conditions in households can lead to prolonged exposure to environmental pathogens, resulting in alterations to the function and morphology of the gut microbiota, which potentially impair children’s growth [15,16,17]. Poor household economic conditions, particularly low income, may exacerbate difficulties in affording nutritious food and accessing healthcare [8]. Stunting can lead to long-term effects such as diminished cognitive function, poor educational achievements, lower productivity, decreased earnings in adulthood, and an increased likelihood of chronic nutrition-related diseases in later life [18]. Despite a steady decline in its incidence, stunting remains a global issue. In 2022, UNICEF, the WHO, and The World Bank reported that 148.1 million children under the age of 5, representing 22.3% of this population, were affected by stunting. Although its incidence continues to decrease, further performance is needed to meet the target prevalence of 13.5% by 2030. The required global average annual reduction (AAR) to reach this target is 6.08%. Most children affected by stunting currently reside in Asia, accounting for 52% of the global total, and Africa, comprising 43% of the global total [19]. Thus, strategies from various sectors are essential to reduce the global burden of stunting. This article discussed the impacts of stunting, including growth failure, cognitive impairments, dysbiosis or enteric electrical dysfunction, endocrine system dysregulation, and anemia, while explaining the mechanisms contributing to these outcomes. Several studies investigating the relationship between the impacts of stunting are summarized in Appendix A, Table A1. Additionally, this article outlines strategies for preventing the impacts of stunting through targeted interventions aimed at reducing both the occurrence and consequences of stunting. A summary of general and specific strategies is presented in Appendix B, Table A2.

## 2. Methods of Search and Inclusion of Articles

Articles were searched on PubMed and Google Scholar, Scopus, and Science Direct, from October 2024 to January 2025. The websites of scientific organizations, such as the WHO, were also searched. The keywords were focused on defining: (1) the impact and consequences of stunting; (2) causes of stunting; (3) growth failure; (4) cognitive impairment; (5) gut microbiota; (6) dysbiosis; (7) anemia; (8) diabetes mellitus; (9) stunting intervention; (10) WASH; (11) SHINE; (12) stunting prevention. All studies were uploaded onto Mendeley’s reference management platform, and identical references were removed.

The screening selection of articles was based on the main topic. The inclusion criteria were (1) classified as cohort study, cross-sectional study, case control study, randomized clinical trials, quasi-experimental study, and observational studies; (2) published between 1 January 2014 and 31 December 2024 in the English language; (3) stunting impact and consequences; (4) cause of stunting; (5) national strategy to accelerate stunting reduction. Exclusion criteria are as follows: articles in a language other than English; in vitro studies; animal studies; historical data; book.

## 3. Stunting Impact and Its Causes

### 3.1. Growth Failure

Nutrition provided to children directly impacts their growth and development from childhood through adulthood [20,21]. A child’s growth is influenced by household conditions, family environment, infections, and dietary patterns [14,20]. Girls with stunting or of short stature (Height-for-Age Z-score < −2) had significantly lower intakes of calcium, β-carotene, riboflavin, niacin, zinc, iron, folic acid, and ascorbic acid than their counterparts with normal height according to a cross-sectional study conducted in India [22]. A meta-analysis revealed that children receiving nutrition enriched with multiple micronutrients experienced a small but significant improvement in height, weight, and motor development. Energy, protein, and other nutrients are essential for promoting growth [23]. The mechanism by which nutritional intake affects growth is illustrated in Figure 2.

As a key macronutrient, protein plays a crucial role in linear growth as it meets the metabolic demands of amino acids necessary for tissue development. Consuming protein can raise the levels of hormones such as insulin and insulin-like growth factor 1 (IGF-1), which promote endochondral ossification and enable longitudinal bone growth [24]. Restriction of protein consumption and reduced energy intake have significantly decreased IGF-1 levels in the body [25,26]. Under hypoglycemia or prolonged fasting conditions, resistance to growth hormone occurs peripherally, accompanied by a reduction in hepatic IGF-1 levels. The decrease in IGF-1 can induce protein catabolism by increasing amino acid levels in the gluconeogenesis pathway [24,27]. A peptide hormone called insulin-like growth factor 1 (IGF-1) promotes cell division, development, and proliferation and is essential for muscle hypertrophy [28,29]. This hormone is primarily synthesized in the liver in response to stimulation by growth hormone (GH) [30]. Apart from this regulating system, the liver and adipose tissue create fibroblast growth factor 21 (FGF-21), essential for metabolism and biological activities related to fatty acid oxidation, gluconeogenesis, and ketogenesis. In fasting states, FGF-21 levels are elevated in the liver, which in turn reduces growth hormone receptor expression and induces growth hormone resistance, as depicted in Figure 3. In addition, IGF-1 may exhibit diurnal fluctuations influenced by growth hormone secretion and feeding cycles. Nutritional deficiency can significantly disrupt the chronobiology (daily rhythmic patterns) of IGF-1 by reducing IGF-1 production, disrupting the growth hormone–IGF-1 axis, altering circadian regulation, and having consequences for growth and development [31,32,33].

Based on the explanation regarding the physiological factors of GH, IGF-I, and FGF21, it can be concluded that adequate nutrition can stimulate the combined actions of growth hormone (GH) and IGF-I during childhood and adolescence to promote long-term growth and somatic maturation. These adaptations may predispose individuals to a heightened sensitivity to IGF-1 oversecretion when exposed to overfeeding or excessive caloric intake in adulthood. The altered metabolic set points, established during childhood, may increase the risk of developing conditions such as obesity and cardiovascular diseases in adulthood, particularly under conditions of caloric excess that stimulate IGF-1 activity [34,35].

### 3.2. Cognitive Impairment

Children experiencing chronic stunting exhibit significantly lower cognitive abilities compared to normal children. Children with stunting are 3.6 times more likely to experience cognitive impairments than children without stunting [36,37]. The first 1000 days of life are a critical time for the growth and development of the brain. Adequate nutrition during pregnancy is essential for the optimal neurodevelopment of the infant. A cross-sectional study conducted in Pakistan found that stunted children exhibited impairments in language abilities and motor adaptation, which is consistent with other research suggesting that malnutrition adversely affects motor skills [38,39]. A diverse range of macronutrients and micronutrients is crucial for brain development. Carbohydrates, as a primary energy source, play a key role in cell metabolism and the formation of brain structures. Protein is involved in developing hippocampal structures, synaptogenesis (particularly essential amino acids), growth factor synthesis, cell proliferation, and differentiation.Additionally, fats are necessary for myelin synthesis, synapse formation, and the visual cortex. Vitamins and minerals, as micronutrients, also contribute significantly to cell metabolism, synaptic function, and myelin formation [40]. Iron plays a critical role in early brain development, particularly during the neonatal period and early childhood. Research has shown that iron deficiency during these stages is a key contributor to disturbances in cognitive development [41,42]. Thus, fats, vitamins, and minerals are indispensable for proper brain growth, especially during the early years. Deficiencies in any of these nutrients can negatively affect cognitive, emotional, and motor development. Balanced nutrition is key to ensuring optimal brain health.

The consequences of stunting on cognitive function can be attributed to inadequate protein intake. Compared to their non-stunted counterparts, children with stunting have reduced levels of critical amino acids. Amino acids are necessary for synthesizing proteins and activating the mammalian target of rapamycin complex 1 (mTORC1) essential for developing numerous tissues, including myelination [43]. When amino acid levels are lowered, mTORC1 disperses in the cytosol and remains inactive. This inactive mTORC1 state may promote lipid synthesis while inhibiting myelination processes within the nervous system, leading to impaired cognition [43,44] (Figure 4).

Stunting can also affect the structure and pathological function of the brain. Chronic malnutrition in the central nervous system can cause tissue damage, reduce synapses and synaptic neurotransmitters, inhibit myelination, and impair the development of dendritic branches during brain development [45]. Stunting and cognitive abilities are strongly associated with lasting effects contributing to poorer educational outcomes over time [46]. Thus, addressing stunting is an urgent public health priority, as early intervention is critical to prevent irreversible cognitive decline. Without timely action, the long-term impact on brain development can hinder a child’s learning potential, academic performance, and future productivity, ultimately affecting both individual well-being and societal progress.

### 3.3. Dysbiosis (Infection, Enteric Environmental Dysfunction)

Stunting is caused by various factors, including intestinal infections and diarrheal diseases [47]. The gut microbiota refers to the collection of microorganisms, primarily bacteria, bacteriophages, fungi, yeasts, and other viruses such as protozoa and archaea, that form a complex ecosystem within the human digestive tract [48,49]. Dysbiosis is an imbalance in the gut microbiota composition caused by chronic consumption of enteric pathogens. These enteric pathogens activate the intestines’ immune system, leading to localized and systemic inflammation. This condition affects the intestines’ physiological and structural integrity, impairing the intestinal barrier’s function and increasing permeability. This state is called enteric environmental dysfunction (EED), which can disrupt nutrient absorption, ultimately resulting in nutrient deficiencies in children [15]; this mechanism is illustrated in Figure 5.

Several studies have shown that children with stunting display a distinctive gut microbiota conformation compared to their non-stunted counterparts. According to a long-term study in India, children with stunting had higher levels of *Desulfovibrio* spp. and lower levels of *Bifidobacterium longum* and *Lactobacillus mucosae* [50]. Stunted children exhibited higher levels of *Escherichia coli*/*Shigella* spp. and *Campylobacter* spp., while non-pathogenic microbiota was found in lower quantities [51]. Moreover, an increase in *Ruminococcus* groups 1 and 2, *Clostridium sensu stricto*, and *Collinsella* was noted in children with stunting, whereas no such changes were observed in non-stunted children [52]. Based on [53], children with stunting have reduced levels of Prevotella 9, which is prevalent in children with appropriate nutritional conditions. Dysbiosis has been associated with reduced short-chain fatty acids (SCFAs) production, increased intestinal permeability, and an elevated susceptibility to infections. Micronutrient metabolism and bioavailability may be impacted by alterations in intestinal barrier function brought on by the microbiome. These procedures may impair intestinal function and restrict children’s ability to grow normally [54]. Zinc homeostasis, the process by which the body maintains a stable zinc level, is primarily regulated through intestinal absorption. A study has demonstrated that a significant portion of dietary zinc intake, approximately 20%, is utilized by the gut microbiota. Another study also revealed that zinc deficiency caused the alteration of the gut microbiota and their functional capacity [55,56]. A global study has shown that diarrheal diseases cause stunting problems in many developing countries [57]. Stunting is also associated with inadequate drinking water sources and poor sanitation facilities. Poor household hygiene practices, such as ineffective handwashing to reduce exposure to enteric pathogens, can lead to diarrhea or inflammation in the digestive tract [58]. In summary, timely intervention in cases of stunting is crucial to prevent dysbiosis, as prolonged undernutrition can disrupt the gut microbiota, weakening immune function and impairing nutrient absorption. Managing stunting early supports healthy growth and development and helps maintain a balanced gut ecosystem essential for long-term health.

### 3.4. Endocrine Dysregulation

#### 3.4.1. Hypothyroid

Iodine is a vital micronutrient necessary for the synthesis of thyroid hormones, thyroxine (T4), and triiodothyronine (T3), which are essential during pregnancy to support the developmental requirements of the fetal nervous system [59]. Iodine needs to increase by over 50% during pregnancy. Insufficient iodine intake during pregnancy can lead to maternal and fetal hypothyroidism. Research has shown the consequences of iodine deficiency based on its timing and severity, with the most severe consequence being cretinism [60]. A deficiency in thyroid hormones, or hypothyroidism, can lead to functional and metabolic disorders that interfere with overall growth and development, a condition known as iodine deficiency disorder (IDD) [61]. The pituitary gland releases thyroid-stimulating hormone (TSH) after receiving a thyrotropin-releasing hormone (TRH) from the brain’s hypothalamus. TSH binds to TSH receptors on thyroid follicular cells. Iodine from consumed food is absorbed into the thyroid gland and stored as iodide (I^−^). The enzyme thyroid peroxidase (TPO) oxidizes this iodide into its active form. The oxidized iodine binds with its precursor protein, thyroglobulin, produced by thyroid follicular cells in a process known as thyroglobulin iodination. This iodination forms monoiodotyrosine (MIT) and diiodotyrosine (DIT). The coupling process then forms the main thyroid hormones, thyroxine (T4) and triiodothyronine (T3). The thyroid gland releases these hormones into the bloodstream, controlling several bodily functions, such as energy metabolism, growth and development, and body temperature regulation [62]. The stages of thyroid hormone formation are depicted in Figure 6.

However, in the case of malnutrition experienced by children with stunting, the regulation of thyroid hormone formation necessary for growth becomes impaired. This represents an adaptive mechanism of endocrine dysregulation, as illustrated in Figure 7. A study reviewing iodine deficiency disorders (IDDs) as a predictor of stunting demonstrated, through a logistic regression model, that students with clinical goiter and urinary iodine concentration (UIC) below 17 µg/L were associated with stunting [63]. Another study revealed that the occurrence of hypothyroidism in infants before birth could affect cognitive abilities, primarily due to hippocampal dysfunction [64]. A study examining the relationship between hypothyroid mice and cognitive function, using mice induced with 5 or 50 ppm propylthiouracil (PTU) in drinking water from day 14 to day 21 post-birth, showed that hypothyroid mice exhibited cognitive impairment. This was evidenced by the OLT (object location test) discrimination ratio, which was significantly altered in the PTU-induced group compared to the control group. Furthermore, this study reported a reduction in neurotransmitter levels (glutamate, *γ*-aminobutyric acid, and glycine) in the hippocampus, as measured by in vivo microdialysis during OLT testing. Thus, the disruption of neurotransmitter secretion can cause hippocampal dysfunction and persist into adulthood [65]. Thyroid hormones play a crucial role in fetal nervous system development. During the mid-first trimester, iodine requirements increased to support T4 hormone production [66]. Total T4, free T4, and T4 binding globulin levels are expected to rise, particularly after the seventh week of pregnancy, facilitating neuronal proliferation and migration in the cerebral cortex. During the second and third trimesters, serum TSH gradually increases, although it remains lower than in non-pregnant women [67].

#### 3.4.2. Diabetes Mellitus

Children with low birth weight are often associated with an increased risk of obesity and type 2 diabetes in adulthood [68,69]. A Japanese study indicates that low birth weight markedly elevates the risk of type 2 diabetes relative to the control group, especially if obesity or overweight occurs in adulthood [70]. Nutritional deficiencies during pregnancy and low birth weight result in the suboptimal development of insulin-producing cells in the pancreas, increasing susceptibility to metabolic disorders, including type 2 diabetes, later in life [71]. A study revealed that children with mild stunting had higher glucose and insulin levels, reduced beta-cell function, and increased insulin resistance as the main characteristics of type 2 diabetes [72]. Under conditions of nutritional deficiency, children experience a reduction in glucose levels, which leads to the decreased secretion of insulin and IGF-1 in the body. Decreased IGF-1 levels can trigger side effects in gluconeogenesis and increased cortisol levels. Additionally, reduced leptin levels can induce increased lipolysis and heightened insulin resistance, hindering glucose absorption in muscle tissues and increasing muscle protein breakdown, leading to elevated cortisol levels in the adrenal glands, as illustrated in Figure 7 [73,74].

#### 3.4.3. Anemia

Anemia, or iron deficiency, is defined as a condition in which the hemoglobin (Hb) levels, relative to body height, are less than 11 g/dL in children under the age of five years. One in four children in Ethiopia experiences Coexisting Anemia and Stunting (CAS), with a prevalence rate of 24.4% from 2005 to 2016 [75]. Anemia in children typically occurs after six months of age and can worsen if iron-rich foods or supplements are not consumed adequately [76]. Fifty percent of anemia cases are attributed to iron deficiency; however, other causes, such as infections, environmental enteropathy, and the spread of infectious diseases, also contribute to the condition. Recurrent infections disrupt nutrient absorption in the digestive tract, leading to nutrient deficiencies and anemia in children [77,78]. Infections may also lead to inflammation, reducing IGF-1 levels, affecting nutrient intake, and increasing the excretion of essential nutrients for growth [9]. Malaria, for instance, induces inflammation and contributes to anemia [79]. Other potential causes of anemia include nutritional deficiencies, such as a lack of vitamins A, B6, B12, C, D, and E, riboflavin, copper, and folic acid. While these deficiencies are relatively rare, and their global contribution to anemia is likely minimal, the impact on anemia development can be synergistic in cases where dietary patterns result in micronutrient deficiencies [80]. Therefore, anemia is considered a multifactorial disease arising from iron deficiency, malaria, helminthiasis, schistosomiasis, hemoglobinopathies, and other micronutrient deficiencies. Additionally, anemia is closely associated with malnutrition-related diseases; conversely, conditions such as stunting, wasting, and obesity can contribute to the development of anemia [81].

## 4. Accelerated Strategies to Reduce Stunting Prevalence as Part of the Sustainable Development Goal (SDG) of Countries

The persistently high prevalence of stunting presents a significant barrier for several countries in achieving the Sustainable Development Goals (SDGs). According to the Conceptual Framework of the Determinants of Child Undernutrition, efforts to reduce stunting prevalence must target both direct and indirect causes of malnutrition. Direct causes are related to insufficient nutritional intake and recurrent infectious diseases. Indirect causes include social conditions (such as infant and child feeding practices, education, sanitation, and workplace conditions), food security (access to adequate nutrition), health environments (access to healthcare services), and household conditions (access to clean water, drinking water, and sanitation) [82,83]. Strategies implemented by various countries to prevent stunting encompass a range of interrelated interventions, including prevention strategies, intervention strategies, and multisectoral policies and approaches, as illustrated in Figure 8.

### 4.1. Prevention Strategies

Effective prevention strategies require addressing the root causes of stunting. Currently, national and international programs, including guidelines issued by the WHO, are available to adopt a comprehensive and multisectoral approach to preventing malnutrition. The WHO recommendations for achieving the Global Nutrition Targets 2025 include several interventions, such as providing energy and protein supplements to pregnant women, implementing programs that educate and promote nutritional needs during pregnancy, monitoring health quality, ensuring attention to hygienic conditions, clean water, and preventing infectious diseases [84].

#### 4.1.1. Maternal Health and Nutrition

Maternal nutritional status during pregnancy is crucial for the child’s early development. The Maternal Nutrition Literacy (MNL) program implemented for pregnant women in Indonesia has proven to have an impact on preventing stunting. During pregnancy, this literacy and education program includes education on exclusive breastfeeding, introducing complementary foods, practical demonstrations on preparing complementary foods, monitoring child growth, and promoting proper sanitation management [85]. The WHO has recommended routine antenatal care (ANC) for pregnant women and adolescent girls through platforms that promote health and prevent diseases. Among these recommendations is the consumption of micronutrients required for pregnant women and adolescent girls, including the use of calcium supplements (1.5–2 g daily) in populations with low calcium intake to prevent preeclampsia, daily iron (30–60 mg) and folic acid (0.4 mg) supplementation to prevent anemia in pregnant women, and vitamin A supplementation (10,000 IU daily or up to 25,000 IU weekly) in populations with a high prevalence of night blindness [86].

In addition, providing essential micronutrients during pregnancy is necessary to prevent stunting. Supplementation programs may include the provision of essential multiple micronutrients such as vitamin A, vitamin B12, zinc, iron, and iodine, as well as the enhancement of staple foods with key nutrients, such as flour, milk, sugar, and cereals, that may assist child growth and development. The consumption of iron–folic acid (IFA) supplements during pregnancy has been shown to significantly impact low birth weight, stunting, and severe stunting in children under the age of 2 years in South Asia [87]. Micronutrient supplementation may also include the provision of lipid-based nutritional supplements (LNS) to provide essential fatty acids and proteins along with micronutrients in quantities of 20–50 grams per day [88,89].

#### 4.1.2. Infant and Young Child Feeding

Infant and young child feeding practices are divided into two key phases: from 0 to 59 months, where exclusive breastfeeding (EBF) is recommended as it serves as protection and strengthens the infant’s immune system; after 6 months, when infants are introduced to complementary foods that are appropriate and rich in essential nutrients, including vitamins and minerals. A study examining the impact of exclusive breastfeeding on children under 2 years old in eastern Indonesia found that children who received exclusive breastfeeding from wealthier households had a 50% lower likelihood of stunting compared to children who were not exclusively breastfed, particularly in poorer households. To strengthen complementary feeding practices, programs are needed to educate caregivers on introducing complementary foods, such as animal-source foods that are rich in nutrients and affordable within the household budget [90].

#### 4.1.3. Improvement of Sanitation and Hygiene

Preventive behaviors to combat stunting through improved sanitation and clean water, as well as the promotion of hygiene practices such as handwashing with soap before meals, are essential in reducing the prevalence of stunting. Campaigns to promote handwashing can prevent diarrheal diseases and other infectious diseases among children [91,92].

### 4.2. Intervention Strategies

#### 4.2.1. Nutritional Intervention

Programs using multivitamins and mineral supplements containing essential nutrients can enhance child growth and development. A study conducted in China revealed that the provision of a supplemental food containing protein, fats, carbohydrates, vitamins A, B1, B2, B12, and D3, folic acid, iron, zinc, and calcium, along with an educational program on complementary feeding, significantly improved the nutritional status of growth-stunted infants [93].

Through a quasi-experimental design, another study conducted in Afghanistan demonstrated the effectiveness of providing specific nutritious foods and implementing a Social and Behavior Change Communication (SBCC) program in preventing growth faltering during the first 1000 days of life for children under 2 years of age. In this program, pregnant and breastfeeding mothers received 7.5 kg of super cereal (250 g/day) during pregnancy and the first 6 months of breastfeeding, while children aged 6–23 months received lipid-based nutrient supplements (LNS), with 30 sachets (50 g/sachet/day) provided every month [94]. Additionally, another study reported that the provision of lipid-based nutrient supplements (LNS) and micronutrient powder (MNP) once daily to children aged 6–23 months in Pakistan led to a reduction in the prevalence of stunting among these children [95].

Specific nutritional interventions are crucial to addressing the consequences of stunting, such as impaired growth and development, dysbiosis or environmental enteric dysfunction (EED), hypothyroidism, and anemia. Specific nutritional strategies that can be implemented to mitigate these issues include the following:

##### Impaired Growth and Development

During periods of growth in children and adolescents, adequate nutrition can stimulate the combined action of growth hormone (GH) and insulin-like growth factor I (IGF-I) to promote long-term growth. In promoting IGF-I activity, providing multiple micronutrients is more effective than single nutrient supplementation in stimulating linear growth. A study in Indonesia on stunted children (aged 48–60 months) with no other hereditary conditions found that supplementing multiple micronutrients, including vitamin A and zinc, significantly increased IGF-1 levels and Z-scores [96].

##### Gut Dysbiosis or Environmental Enteric Dysfunction (EED)

Efforts to address gut dysbiosis in the context of stunting can be approached through three key interventions: an application of probiotics, prebiotics, and synbiotics, as illustrated in Figure 9.

Probiotics provide advantages to the host when administered in sufficient quantities. They can stimulate child growth by modulating the gut microbiota and immune system, inhibiting pathogen growth, preventing infections, and enhancing nutrient absorption. Meanwhile, prebiotics strengthen the growth of specific bacteria, particularly *bifidobacteria* and *lactobacilli*, which modulate short-chain fatty acid (SCFA) production to inhibit pathogen growth [97]. A study conducted by Barratt et al., 2022 in Bangladesh with infants aged 2–6 months receiving probiotics in the form of B. infantis EVC001 single-strain or in combination with human milk oligosaccharide (LNnT) at a dose of 8 billion CFU per day showed significant increases in weight gain and a reduction in inflammation in the digestive tract [98]. Synbiotics, a combination of prebiotics and probiotics explicitly formulated to stimulate the effects of probiotic organisms, have also shown benefits. For instance, administering synbiotics (B. infantis EVC001 combined with lacto-N-neotetraose [LNnT]) led to increased weight compared to a placebo group [99]. Children with a history of exclusive breastfeeding have more *Bifidobacterium* and *Odoribacter splanchnicus* bacteria than those not exclusively breastfed [92]. In addition to preventing and managing diarrhea in children, improving immunization coverage can offer better protection. To improve and maintain a balanced gut microbiota composition, children should consume healthy fermented foods and a proportionate intake of fiber, fats, carbohydrates, and proteins [16].

##### Hypothyroidism

Adequate iodine intake is necessary to maintain normal thyroid function. The WHO recommends iodine intake levels for infants and children to be 90–120 µg/L daily. Breast milk is a key source of iodine for infants in the first six months of life. However, the mother’s dietary iodine intake directly influences the iodine content. Therefore, ensuring adequate iodine nutrition in pregnant and lactating women is critical not only for the mother’s health but also for the growth and development of the infant [100]. In addition to iodine, other nutrients such as iron, vitamin D, zinc, and other micronutrients also play a role in thyroid function. Iron deficiency, for instance, can reduce thyroid peroxidase activity, an enzyme essential for thyroid hormone production. Thus, integrated nutrition interventions that address multiple micronutrient deficiencies are more effective in preventing thyroid-related growth delays [101]. A study conducted in China revealed that the provision of a supplementary food product (hibao) containing protein, fat, carbohydrates, vitamins A, B1, B2, B12, D3, folic acid, iron, zinc, and calcium, combined with an educational program on complementary feeding practices, significantly improved the nutritional status of infants experiencing growth faltering [93].

##### Anemia

Currently, both national and international programs are available, including guidelines issued by the WHO to adopt a comprehensive and multisectoral approach to malnutrition prevention. Micronutrient supplementation has been shown to significantly reduce the prevalence of anemia, especially in vulnerable groups such as pregnant women, infants, and young children, thereby contributing to improved health outcomes and nutritional status [84]. A study conducted in Pakistan in 2023 demonstrated that the provision of Wawamum LNS-MQ (lipid-based nutrient supplement—medium quantity) could increase hemoglobin concentration, micronutrient status, and other growth parameters in children aged 6 to 23 months. This intervention can be enhanced as a form of malnutrition prevention in Pakistan and other developing countries [102]. Another evidence comes from the administration of Yingnyangbao, which contains energy, protein, vitamins A, D, B1, B2, B12, folic acid, calcium, iron, and zinc, has been shown to effectively increase hemoglobin concentration, reduce the risk of anemia, and improve nutritional status, thereby lowering the prevalence of stunting [101]. Stewart et al. (2020) conducted a Multiarm Cluster-Randomized Controlled Trial involving 125 communities, which revealed that the group of children aged 6 to 18 months and pregnant women through children aged 6 months who received lipid-based nutrient supplements exhibited a 25% lower prevalence of anemia and iron deficiency compared to the control group [103].

#### 4.2.2. Early Childhood Development Programs

These programs include education and family support to stimulate early cognitive and physical development, which can help mitigate the consequences of stunting. Early childhood development programs must also be supported by various initiatives that educate and promote the nutritional needs of pregnant women, nutrition assessments, immunization campaigns, monitoring of child growth, ensuring proper hygiene, providing access to clean water, and preventing infectious disease [84]. Moreover, the family component of ECD programs complements health efforts by promoting a nurturing and stimulating home environment essential for healthy child development. This support may come in the form of parenting education, nutritional guidance, and access to social services that ensure families have the resources to provide adequate care [104].

#### 4.2.3. Strengthening the Health System

Strengthening the health system aims to provide optimal and comprehensive healthcare services for mothers and children, including regular check-ups, immunization, and growth monitoring. The WHO has explained the six building blocks of a health systems, which are the following: (1) health services that are effective, safe, and efficient [105]; (2) health workforce that are sufficient, well-distributed, skilled, and responsive; (3) health information systems that produce and use reliable data for decision making; (4) access to essential medicines and technologies that are of high quality, safe, and cost-effective; (5) health financing systems that ensure equitable access; (6) leadership and governance that provide strong policy frameworks, oversight, and accountability As part of Indonesia’s national strategy to accelerate the prevention of child stunting, several programs have been introduced to improve access to and the quality of nutrition and healthcare services. These include the provision of access to the National Health Insurance (JKN), access to family planning (FP) services, and conditional cash transfer programs for low-income families, such as the Family Hope Program (PKH) [82].

### 4.3. Multisectoral Policies and Approaches

Stunting is a complex issue that requires coordinated solutions across multiple sectors. Effective policies that can be implemented include:

#### 4.3.1. Local and National Policies

Support from local and national policies is crucial, integrating nutrition into broader policies across education, agriculture, health, and social protection sectors. Several integrated interventions have been implemented, such as the WASH (Water, Sanitation, Hygiene) program in Bangladesh and Kenya, the SHINE (Sanitation, Hygiene, Infant Nutrition Efficacy) trial program in Zimbabwe, and the STRANAS (National Strategy to Accelerate Stunting Reduction) program in Indonesia. Multisectoral policy interventions have a significant impact on reducing the prevalence of stunting worldwide [82,90].

#### 4.3.2. Monitoring and Evaluation

Monitoring and evaluation focus on assessing the results achieved, including the program’s impact and outcomes, identifying any discrepancies during implementation and monitoring factors that can accelerate the prevention of stunting [82].

## 5. Limitations and Future Research

There are some limitations to our narrative review. In this study, only four databases and Mendeley were used as access to find the sources of the articles being reviewed. A considerable proportion of the reviewed literature, particularly that from the Asian region, predominantly comprised studies conducted in Indonesia. To gain a more comprehensive understanding of the factors and conditions linked to stunting, it would be beneficial to include more research from a broader range of countries, especially in countries with a high prevalence of stunting.

As authors, we made every effort to ensure that the studies we collected were as objective as possible, as may be indicated by, among other things, the large number of studies cited and the reporting of all study results. However, despite the authors’ efforts, a narrative review is a form of scientific work that may carry a greater risk of subjective evaluation than other review works, such as a systematic review or meta-analysis.

In future studies, there is a need for long-term monitoring of both prevention and intervention programs. Longitudinal studies and implementation research across different policy and health system contexts will be particularly valuable in informing sustainable, evidence-based strategies to meet global nutrition targets, including the SDGs.

## 6. Conclusions

This review has explored the significant global burden of stunting and its short- and long-term consequences, including impaired physical growth, cognitive delays, dysbiosis, endocrine dysfunction, and anemia. The evidence suggests that early interventions, particularly those targeting maternal nutrition and child health in the first 1000 days of life, are critical to preventing permanent growth failure and developmental setbacks. This also raises attention to which strategies that may accelerate the reduction of stunting incidence by strengthening three following aspects: (1) Prevention Strategies, including maternal health and nutrition, promotion of exclusive breastfeeding and appropriate complementary feeding, as well as improved sanitation and hygiene practices; (2) Intervention Strategies, such as micronutrient supplementation, early childhood development programs, and the strengthening of healthcare systems to ensure continuity and quality of care for mothers and children; (3) Multisectoral Policies and Approaches emphasizing the integration of nutrition into broader policy frameworks at both local and national levels, and the importance of robust monitoring and evaluation systems. Looking forward, further research and action are needed in several key areas: (1) Future programs should be adapted to local sociocultural and economic realities, especially in high-burden regions, through participatory approaches involving community members and local stakeholders; (2) more longitudinal research is needed to evaluate the lasting effects of stunting interventions, particularly those targeting early development and maternal care; (3) governments and international partners must sustain political will, policy alignment, and funding to support multisectoral strategies that aim not only to treat but to prevent stunting at its roots.

## Figures and Tables

**Figure 1 nutrients-17-01493-f001:**
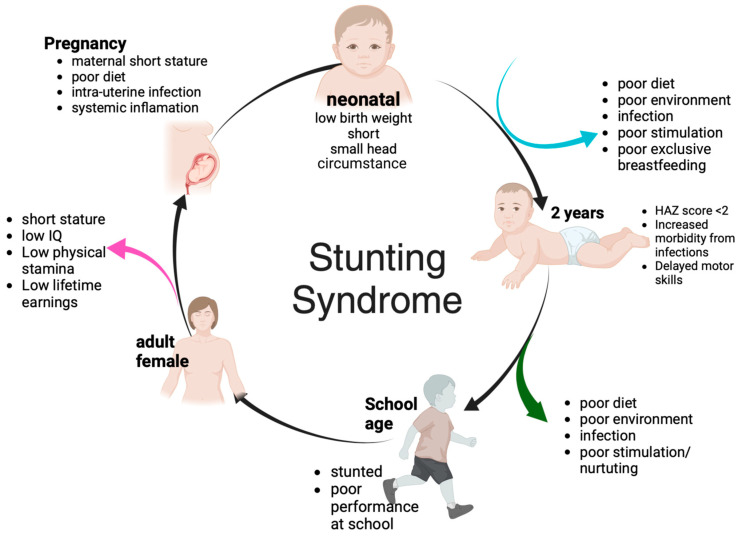
Stunting Syndrome: This figure explains how stunting syndrome is identified as a continuous cycle. Several factors that influence the intergenerational stunting syndrome cycle include genetic factors, nutritional deficiencies in mothers and children throughout life, inadequate breastfeeding, improper complementary feeding, infectious diseases and inflammation, and social factors such as lack of resources across generations and poverty. Various pathological alterations due to impaired linear growth in early life are associated with heightened morbidity and mortality rates, decreased physical ability, poor brain development, diminished economic prospects, and an increased risk of metabolic disorders in adulthood. HAZ—Height-for-Age Score (Created in BioRender, https://www.biorender.com/, accessed on 18 April 2025).

**Figure 2 nutrients-17-01493-f002:**
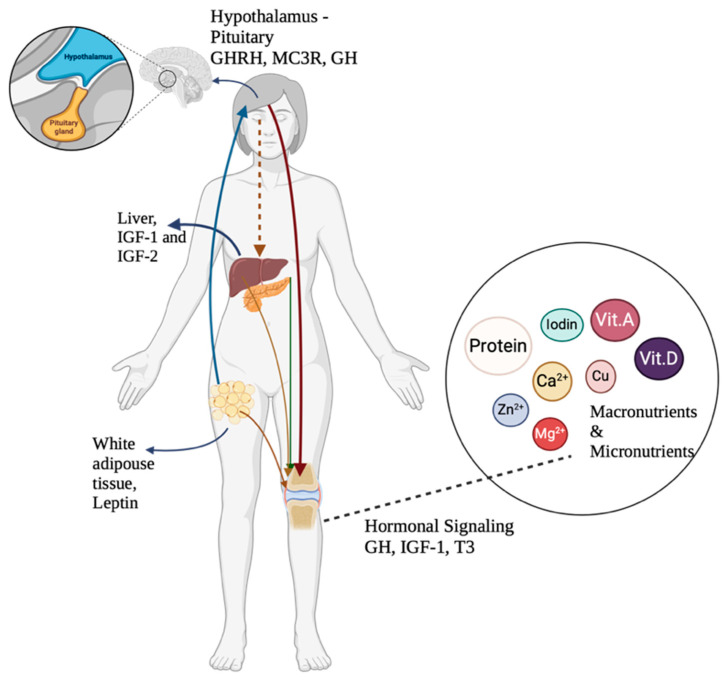
Explanation of the role of nutrition and endocrine regulation in stimulating growth. IGF-1—insulin-like growth factor 1; IGF-2—insulin-like growth factor 2; GH—growth hormone; GHRH—Growth Hormone Releasing Hormone; MC3R—Melanoconine-3 Receptor; T3—triiodothyronine; Ca^2+^—calcium; Zn^2+^—zinc; Mg^2+^—magnesium; Cu—copper (Created in BioRender, https://www.biorender.com/, accessed on 3 December 2024).

**Figure 3 nutrients-17-01493-f003:**
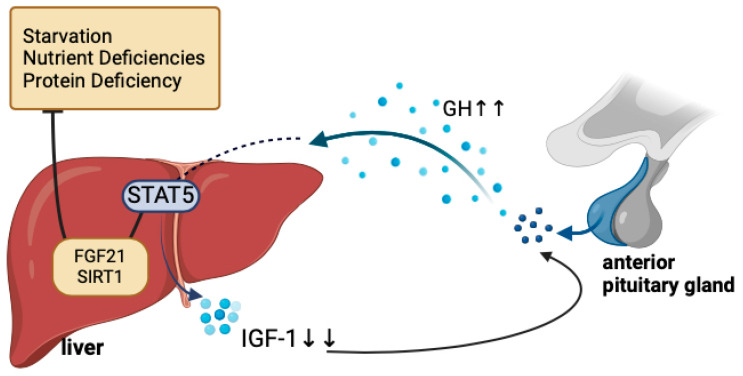
Mechanism of how fasting, nutrient deficiency, and protein deficiency affect growth hormones (GH, IGF-1, FGF21). IGF-1—insulin-like growth factor 1; GH—growth hormone; ↑—increase; ↓—decrease (Created in BioRender, https://www.biorender.com/, accessed on 23 December 2024).

**Figure 4 nutrients-17-01493-f004:**
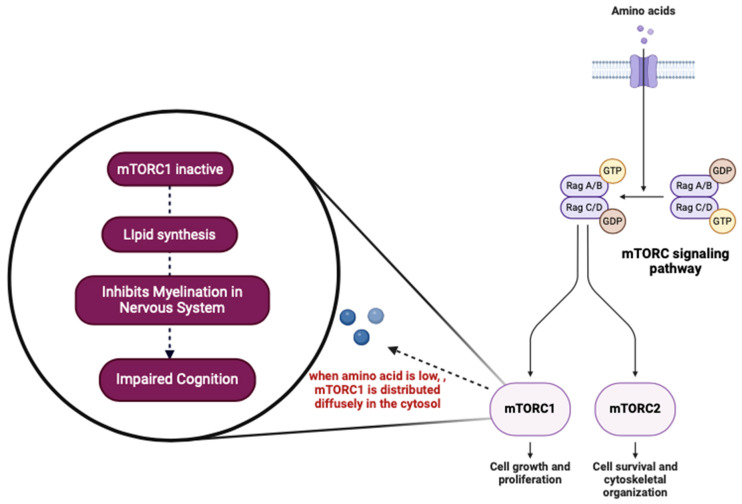
The role of amino acids in the mTORC pathway in inducing impaired cognition. mTORC—mammalian target of rapamycin; mTORC1—mammalian target of rapamycin complex 1; mTORC2—mammalian target of rapamycin complex 2; GDP—guanosine diphosphate; GTP—guanosine triphosphate; Rag—regulator proteins, Rag A/B is small rags, Rag C/D is large rags, while Rag A/B forms a heterodimer with product of Rafg C/D (Created in BioRender, https://www.biorender.com/, accessed on 23 December 2024).

**Figure 5 nutrients-17-01493-f005:**
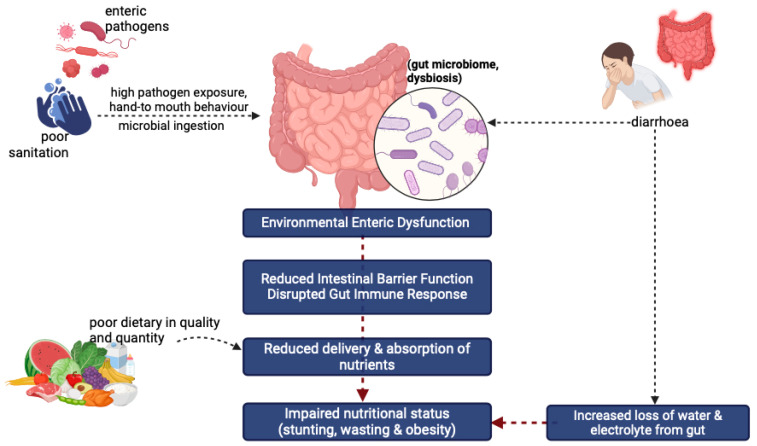
Pathophysiology EED induced impaired nutritional status (Created in BioRender, https://www.biorender.com/, accessed on 24 December 2024).

**Figure 6 nutrients-17-01493-f006:**
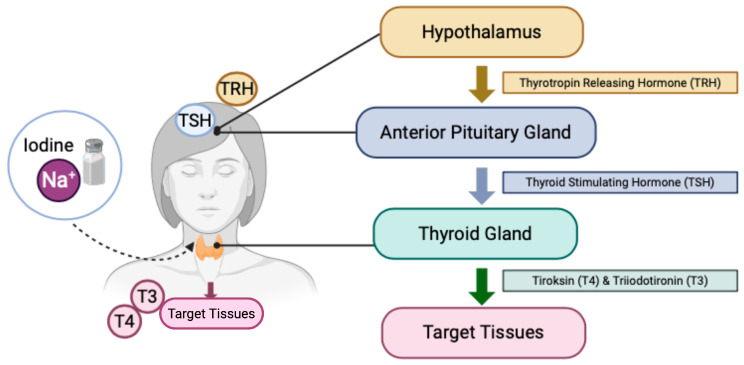
Mechanism of thyroid hormone synthesis. TSH—thyroid-stimulating hormone; TRH—thyrotropin-releasing hormone; T4—thyroxine; T3—triiodothyronine (Created in BioRender, https://www.biorender.com/, accessed on 30 December 2024).

**Figure 7 nutrients-17-01493-f007:**
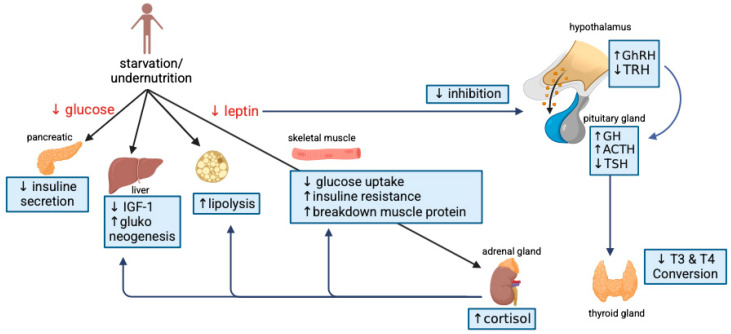
Endocrine dysregulation during starvation and undernutrition. IGF-1—insulin-like growth factor 1; GH—growth hormone; GhRH—growth hormone-releasing hormone; TSH—thyroid-stimulating hormone; TRH—thyrotropin-releasing hormone; ACTH—adrenocorticotropic hormone; T4—thyroxine; T3—triiodothyronine; ↑—increase; ↓—decrease (Created in BioRender, https://www.biorender.com/, accessed on 18 January 2025).

**Figure 8 nutrients-17-01493-f008:**
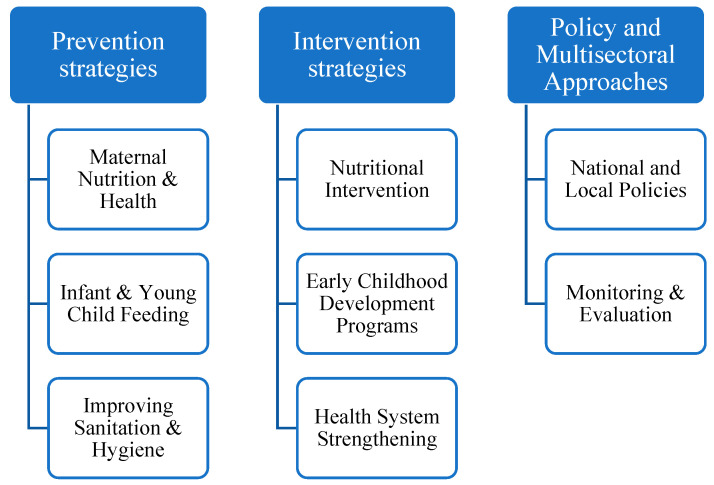
Multisectoral strategies for reducing stunting prevalence. (Created on Microsoft Word, https://www.microsoft.com/en-us/microsoft-365/word, accessed on 30 December 2024).

**Figure 9 nutrients-17-01493-f009:**
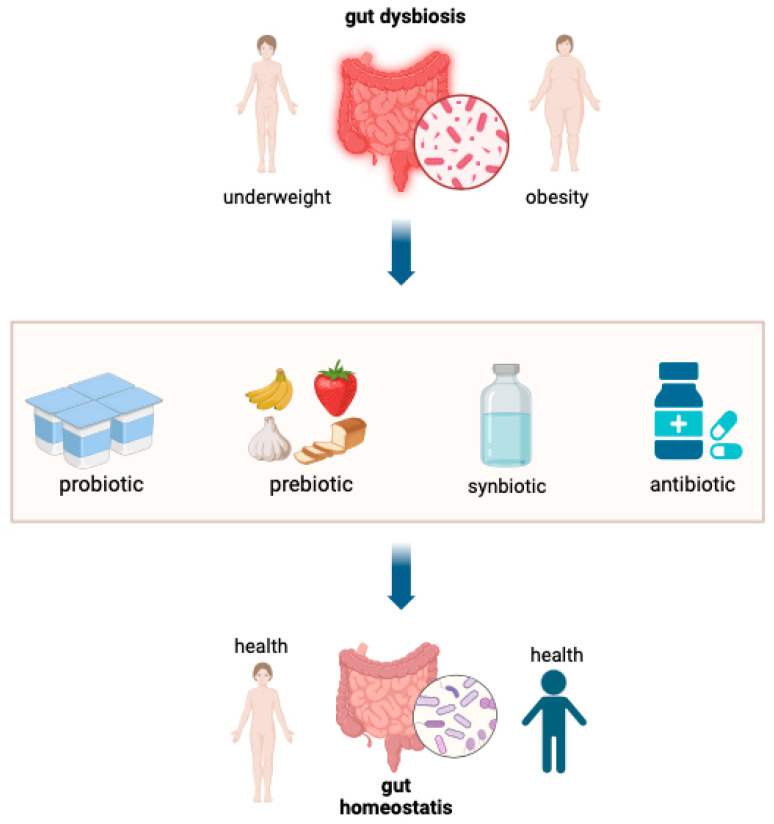
Scheme for treating gut dysbiosis in addressing stunting (Created in BioRender, https://www.biorender.com/, accessed on 24 December 2024).

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
