# Peer review of "Understanding Stunting: Impact, Causes, and Strategy to Accelerate Stunting Reduction—A Narrative Review"

_nutrients, 2025, doi:10.3390/nu17091493_

Round 1

Reviewer 1 Report

Comments and Suggestions for Authors

I have serious concerns regarding this manuscript’s publications. Apart from the lack of novelty, the search methods are not proper.

The abstract is not adequate. It should be adequately structured according to the journal’s instructions and provide the most relevant data analyzed in the conducted work.

The Introduction is fine. However, the Methods are not adequate. More data should be included, like the keywords used in the search, and the inclusion/exclusion criteria. I don’t agree with the databases where you performed the search. Why only these 2 databases? PubMed is very specific to medical sciences, and Google Scholar is grey literature. You should include more adequate databases in your search like Scopus, Web of Science, Proquest… This is a relevant limitation and, in my opinion, it prevents the publication of your study in its current state.

Do you have the copyrights of the presented figures? Please mention this clearly in the manuscript.

The topics discussed on page 14 should be explained in detail. The provided information is scarce.

Study limitations are missing.

The Conclusions are nothing but a summary and it is not adequate. Future perspectives are missing.

Reviewer 2 Report

Comments and Suggestions for Authors

I have read this paper with interest. As reflected in the title and abstract, the paper is a narrative review on stunting, with focus on the impact, causes and potential strategies to accelerate the reduction in stunting incidence. I have provided my comments and suggestions sequentially, not necessarily reflected the ‘majority’ of these comments.

In the appendix table A1 and B1, you suggest to summarize all impacts and causes, but since this is not a systematic review, I would suggest to remove ‘all’.

Figure 1: textual, circumtance, exclusife, syndrome, … in maturity please check, and why only adult female, were is the male adult in your construct ? (and for figure 1 and figure 2, , 3 and subsequent, abbreviations in the figures should be explained in the legend.

In the introduction, it is not always clear if we target prenatal versus postnatal interventions. In my reading, there is likely a need to further add and comment on pregnancy related interventions, while some factors relate to both the pre- and postnatal setting. In the ‘prevention section’,4 this is much clearer.

I value the use of a method section, but suggest to add the search period or day, and add some clarity on the search terms applied.

Line 110: linear development, what do you mean with these words ?

Related to Figure 3: is there anything known on how nutritional deficiency affects the ‘chrono-biology’ of IGF 1 (cyclic over the day).

Line 128-134: Related to the Barker hypothesis, stunting in early life makes the adult likely more

sensitive to IGF-1 oversecretion when overfed ? Please consider to add some lines on this

vulnerability.

On the cognition alinea, there is likely some value to add some information on iron ?

On the dysbiosis, there is likely some value to add some lines on zinc ? what about parasites or

deworming strategies ?

Hypothirioid: please check textual. Is there any value to add some sentences on the impact of

iodine deficienty during pregnancy ?

On diabetes, do you mainly refer to type 2 diabetes ? please be more explicit on this. 

Round 2

Reviewer 1 Report

Comments and Suggestions for Authors

I congratulate the authors for bringing this manuscript to an acceptable level.

Reviewer 2 Report

Comments and Suggestions for Authors

as a narrative review, there are some a priori priority issues, while the authors have replied accurate to my suggestions.